# Regulation of cell protrusions by small GTPases during fusion of the neural folds

Ana Rolo[1]*, Dawn Savery[1], Sarah Escuin[1], Sandra C de Castro[1], Hannah EJ Armer[2], Peter MG Munro[2], Matteo A Molè[1], Nicholas DE Greene[1], Andrew J Copp[1]*

[1]Newlife Birth Defects Research Centre, Institute of Child Health, University College London, London, United Kingdom; [2]Imaging Unit, Institute of Ophthalmology, University College London, London, United Kingdom

**Abstract** Epithelial fusion is a crucial process in embryonic development, and its failure underlies several clinically important birth defects. For example, failure of neural fold fusion during neurulation leads to open neural tube defects including spina bifida. Using mouse embryos, we show that cell protrusions emanating from the apposed neural fold tips, at the interface between the neuroepithelium and the surface ectoderm, are required for completion of neural tube closure. By genetically ablating the cytoskeletal regulators Rac1 or Cdc42 in the dorsal neuroepithelium, or in the surface ectoderm, we show that these protrusions originate from surface ectodermal cells and that Rac1 is necessary for the formation of membrane ruffles which typify late closure stages, whereas Cdc42 is required for the predominance of filopodia in early neurulation. This study provides evidence for the essential role and molecular regulation of membrane protrusions prior to fusion of a key organ primordium in mammalian development.

*For correspondence: a.rolo@ucl.ac.uk (AR); a.copp@ucl.ac.uk (AJC)

**Competing interests:** The authors declare that no competing interests exist.

## Introduction

The fusion of apposed epithelial sheets is an essential process in the completion of many morphogenetic events including closure of the neural tube, optic fissure, palatal shelves, and cardiac septa. Failure of these fusion events leads to clinically important congenital malformations including neural tube defects (NTDs: anencephaly and open spina bifida), coloboma, cleft palate, and cardiac septal defects, respectively (*Pai et al., 2012*; *Ray and Niswander, 2012*).

NTDs are among the commonest human birth defects, affecting 0.5–2 per 1000 pregnancies worldwide (*Copp et al., 2013*). Understanding the mechanisms by which the vertebrate neural plate folds up and fuses to form a closed neural tube is thus of paramount importance for gaining insight into the embryonic pathogenesis of NTDs, and for developing improved methods for their prevention. In recent years, some of the molecular mechanisms underlying different morphogenetic aspects of neural tube closure have been unravelled. For example, the initial convergence and extension movements that narrow and elongate the neural plate were found to be regulated by the non-canonical Wnt-planar cell polarity pathway (*Wallingford and Harland, 2002*; *Williams et al., 2014*; *Ybot-Gonzalez et al., 2007b*), whereas the subsequent bending of the mammalian neural plate at discrete medial and dorsolateral hinge points was shown to be regulated by Shh and BMP signalling (*Ybot-Gonzalez et al., 2002*; *Ybot-Gonzalez et al., 2007a*). Much less is known, however, about the final steps of neurulation, involving fusion and remodelling of the neural folds at the dorsal midline.

During epithelial 'fusion', individual cells do not actually fuse with one another, but rather cells at the leading edges of apposed tissues form de novo adhesions to create a continuous epithelium. Neural tube closure involves a particular kind of epithelial fusion, in which two distinct tissues need to fuse and remodel: the pseudostratified neuroepithelium (NE) and the squamous surface ectoderm

**eLife digest** The neural tube is an embryonic structure that gives rise to the brain and spinal cord. It originates from a flat sheet of cells – the neural plate – that rolls up and fuses to form a tube during development. If this closure fails, it leads to birth defects such as spina bifida, a condition that causes severe disability because babies are born with an exposed and damaged spinal cord.

As the edges of the neural plate meet, they need to fuse together to produce a closed tube. It was known that cells at these edges extend protrusions. However, it was unclear how these protrusions are regulated, whether they arise from neural or non-neural cells and whether or not they are required for the neural tube to close fully.

By studying mutant mouse embryos, Rolo et al. found that cellular protrusions are indeed required for the neural tube to close completely. These protrusions proved to be regulated by proteins called Rac1 and Cdc42, which control the filaments inside the cell that are responsible for cell shape and movement. Rolo et al. also found that the cells that give rise to the protrusions are not part of the neural plate itself. Instead, these cells are neighboring cells from the layer that later forms the epidermis of the skin (the surface ectoderm).

Future studies will need to investigate which signals instruct those precise cells to make protrusions and to discover what happens to the protrusions after contact is made with cells on the opposite side. It will also be important to determine whether spina bifida may arise in humans if the protrusions are defective or absent.

(SE). Initially, these two tissues form a continuous ectodermal layer; however, during neural fold fusion, the continuity of this epithelium is disrupted at the bilateral NE/SE junctions, and new adhesions form between concurring epithelia from each side. Remodelling then generates the closed neural tube covered by the future epidermis (*Figure 1*). Cellular protrusions are often observed prior to apposition at the onset of epithelial fusion events (*Pai et al., 2012*). It has long been known that membrane ruffles are present at the edges of apposed neural folds during neural tube closure in amphibians (*Mak, 1978*), birds (*Bancroft and Bellairs, 1975*; *Schoenwolf, 1979*), and mammals (*Geelen and Langman, 1979*; *Waterman, 1976*) (*Figure 1*). Recently, filopodia have been observed at the neural fold tips in mice (*Massarwa and Niswander, 2013*; *Pyrgaki et al., 2010*) and in ascidians (*Ogura et al., 2011*), together with F-actin enrichment along the NE/SE boundary (*Hashimoto et al., 2015*; *Ogura et al., 2011*). Early morphological studies in mice found that the initial contact between neural folds in the midbrain and hindbrain regions is made by SE cells, from which cellular protrusions emanate, whereas at the forebrain level initial contact is made by NE cells (*Geelen and Langman, 1977*; *1979*). In chick, during cranial neurulation the SE and NE layers contact at the same time, but in the spinal region this first contact is made by SE cells (*Schoenwolf, 1979*), and in frog neurulation the SE closes first, and this closure is actually uncoupled from NE closure, which occurs later (*Davidson and Keller, 1999*). In mouse spinal neural tube closure, however, the cell type of origin of the protrusive cells and initial contact point have not been previously identified. Moreover, whether these protrusions are required for vertebrate neural tube closure at any level of the body axis is unknown.

Small GTPases of the Rho family are ubiquitously expressed molecular switches that cycle between active (GTP-bound) and inactive (GDP-bound) states and have a pivotal role in linking extracellular signals with several specific downstream effectors. In particular, Rac1 and Cdc42 are well-known regulators of the actin cytoskeleton that drives cellular protrusion. Rac1 induces the formation of the branched actin networks that underlie lamellipodia and membrane ruffles, and Cdc42 drives the assembly of unbranched actin bundles that form filopodia (*Heasman and Ridley, 2008*; *Ridley, 2011*). Even though the specific roles of the different Rho-GTPases were initially described using constitutively active and/or dominant negative forms, as well as pharmacological approaches, these techniques were later recognised to create limitations, owing to issues with specificity and dosage control. Conditional gene targeting has subsequently become the preferred method for studying Rho GTPase function in vivo in mammals, particularly mice (*Wang and Zheng, 2007*).

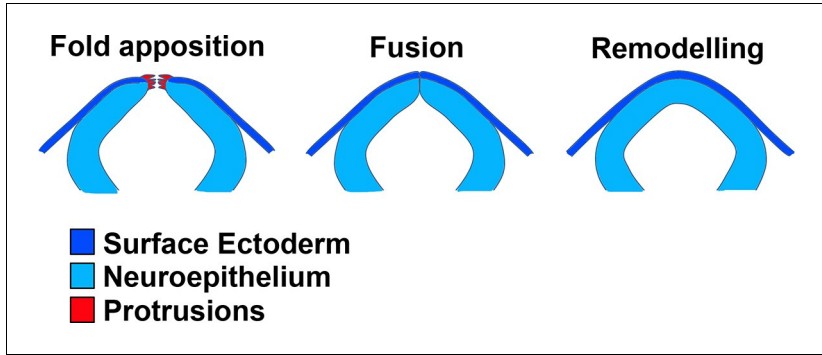

**Figure 1.** Schematic representation of the final events of neurulation in the spinal region of the mouse embryo. The apposing neural folds exhibit cell protrusions from their tips (left), the neural folds then undergo fusion (middle), and the two epithelia remodel to generate a closed neural tube covered by SE (right). DOI: 10.7554/eLife.13273.003

Here, we describe the formation of different types of protrusions at the edges of the mouse spinal neural folds immediately prior to fusion. Using conditional targeting of Rac1 or Cdc42 in the NE and/or the SE, we show that these protrusions originate from SE, rather than NE cells. Furthermore, we show that Rac1 regulates the formation of ruffles, without which neurulation fails leading to open spina bifida, whereas Cdc42 is implicated in the formation of filopodia during earlier stages of neurulation.

## Results

### Varying morphology of cell protrusions as spinal neural tube closure progresses

Membrane protrusions at the tips of the mouse neural folds have been described using both transmission electronic microscopy (TEM) (*Geelen and Langman, 1979*) and scanning electronic microscopy (SEM) (*Waterman, 1976*). TEM provides very detailed imaging of both the cellular protrusions and the cells they emanate from. However, the sectional views obtained with TEM do not allow for a three-dimensional analysis of protrusive morphology. We therefore initially chose to describe the protrusive activity in the mouse spinal neural folds using SEM. We observed elaborate membrane protrusions at the point of fold apposition throughout spinal neurulation, and these protrusions were found to vary qualitatively as neurulation progressed (*Figure 2*). At the onset of neural tube closure (somite-stage (ss)7), protrusions consisted mainly of long, finger-like filopodia (*Figure 2A*). As neurulation progressed (e.g. ss12), we observed a mixture of filopodia and ruffles at the spinal fusion point (*Figure 2B*). By late spinal neurulation stages (from ss24 onwards), membrane ruffles only were observed, devoid of filopodial extensions (*Figure 2C*). In some cases, cell protrusions were seen not only at the fold apposition point, but also along the edges of both open neural folds (*Figure 2C* and *Figure 2—figure supplement 1*). For consistency in the analysis between different embryos, we have focussed our analysis of protrusion types solely on the activity seen at the fold apposition point (for further details of protrusive analysis, see 'Materials and methods' section and *Figure 2—figure supplements 2* and *3*).

### Targeting Rho GTPases in the NE and SE during spinal neurulation

Rac1 and Cdc42 are small Rho GTPases that regulate the cytoskeleton, particularly the actin networks that underlie the formation of lamellipodia/ruffles and filopodia, respectively (*Heasman and Ridley, 2008*; *Ridley, 2011*). Knock-outs of both *Rac1* and *Cdc42* are embryonic lethal before neurulation (*Chen et al., 2000*; *Sugihara et al., 1998*), and therefore, analysing their role in neural tube closure required the generation of conditional knock-out mice. We initially chose to conditionally ablate these GTPases by recombining floxed alleles of either *Rac1* or *Cdc42* with Cre recombinase expressed under the control of the *Pax3* promoter. Pax3 is a transcription factor expressed in the dorsal-most cells of the developing neural plate and neural tube from early neurulation stages

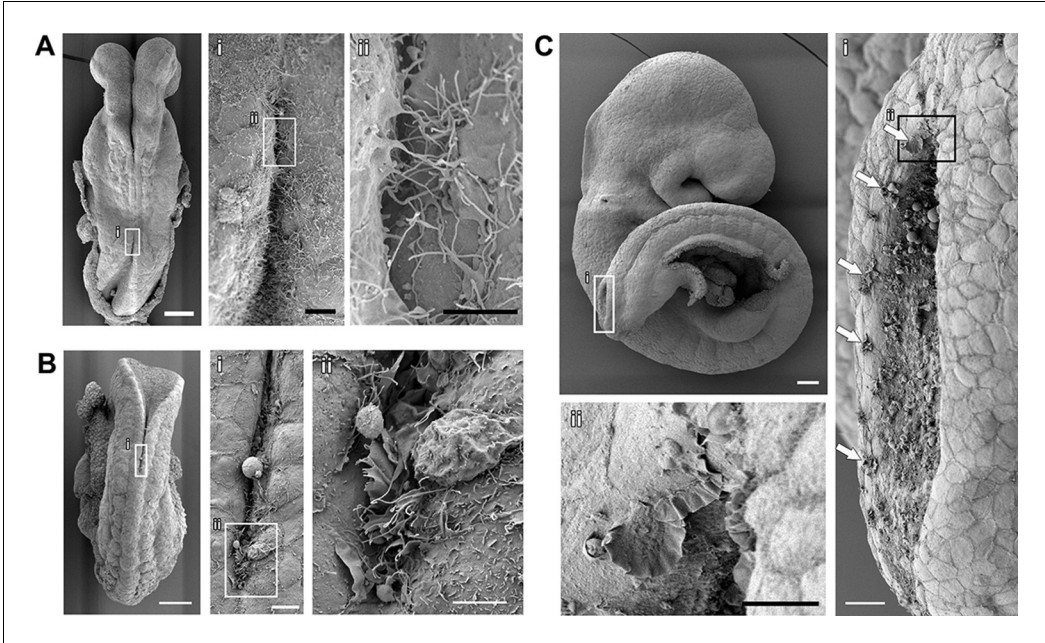

**Figure 2.** Cell protrusions emanate from the interface between the NE and the SE of apposing neural folds during spinal neurulation. (**A–C**) SEMs of ss7 (**A**), ss12 (**B**), and ss24 (**C**) embryos. The point of spinal neural fold apposition exhibits filopodia at ss7, filopodia and ruffles at ss12 and membrane ruffles at ss24. Note the presence of ruffles at intervals along the edges of the PNP neural folds (Ci, arrows). At least 10 different embryos were analysed. Scale bars: 100 μm (**A**, **B**, **C**), 20 μm (Ai, Ci), 10 μm (Aii, Bi, Cii), and 5 μm (Bii).

The following figure supplements are available for figure 2:

**Figure supplement 1.** Protrusions are present on the tips of both neural folds.

**Figure supplement 2.** Examples of different types of protrusions visualized by SEM at the PNP point of fusion.

**Figure supplement 3.** Quantification of the types of protrusion observed in all control embryos used in the different crosses in this study.

(embryonic day (E)8.5) (*Goulding et al., 1991* and *Figure 3—figure supplement 1*). To confirm effective Cre-driven recombination at the appropriate tissues and stages, we crossed $Pax3^{Cre/+}$ mice (*Engleka et al., 2005*) with a homozygous ROSA26-EYFP reporter line (*Srinivas et al., 2001*). As expected, YFP was expressed in the dorsal NE from E8.5 onwards (*Figure 3A,B*), with some YFP-expressing cells also detected ventral to the Pax3 expression domain, consistent with recent findings (*Moore et al., 2013*). Surprisingly, however, at neurulation stages later than ss20, we also detected YFP expression in cells of the dorsal SE, mainly those directly in contact with the NE of the open neural folds (*Figure 3B*). In confirmation of their SE identity, we found that these cells robustly express E-cadherin, whereas Pax3 was expressed only at very low intensity, or not at all (*Figure 3—figure supplement 1*).

## Rac1 is required for the late stages of spinal neural tube closure

When *Rac1* was ablated in the Pax3 lineage (confirmed by mRNA in situ hybridisation, see *Figure 3—figure supplement 2*), 76% of embryos displayed spinal NTDs consisting of either open spina bifida or a curled tail (*Figure 3C*; *Table 1*). These defects occurred at a similar frequency in both $Pax3^{Cre/+}$; $Rac1^{flox/-}$ (21/27) and $Pax3^{Cre/+}$; $Rac1^{flox/flox}$ (16/22) embryos (p=0.947), and hence these genotypes were combined for further analysis (denoted Pax3Cre-Rac1). In contrast, $Pax3^{Cre/+}$; $Rac1^{flox/+}$ and $Pax3^{Cre/+}$; $Rac1^{+/-}$ control embryos (denoted Pax3Cre-Con), which had conditional or constitutional heterozygous *Rac1* loss of function, exhibited only 8% spina bifida, a significantly lower frequency than in Pax3Cre-Rac1 embryos (*Figure 3C*; *Table 1*). The third genotype group comprised embryos

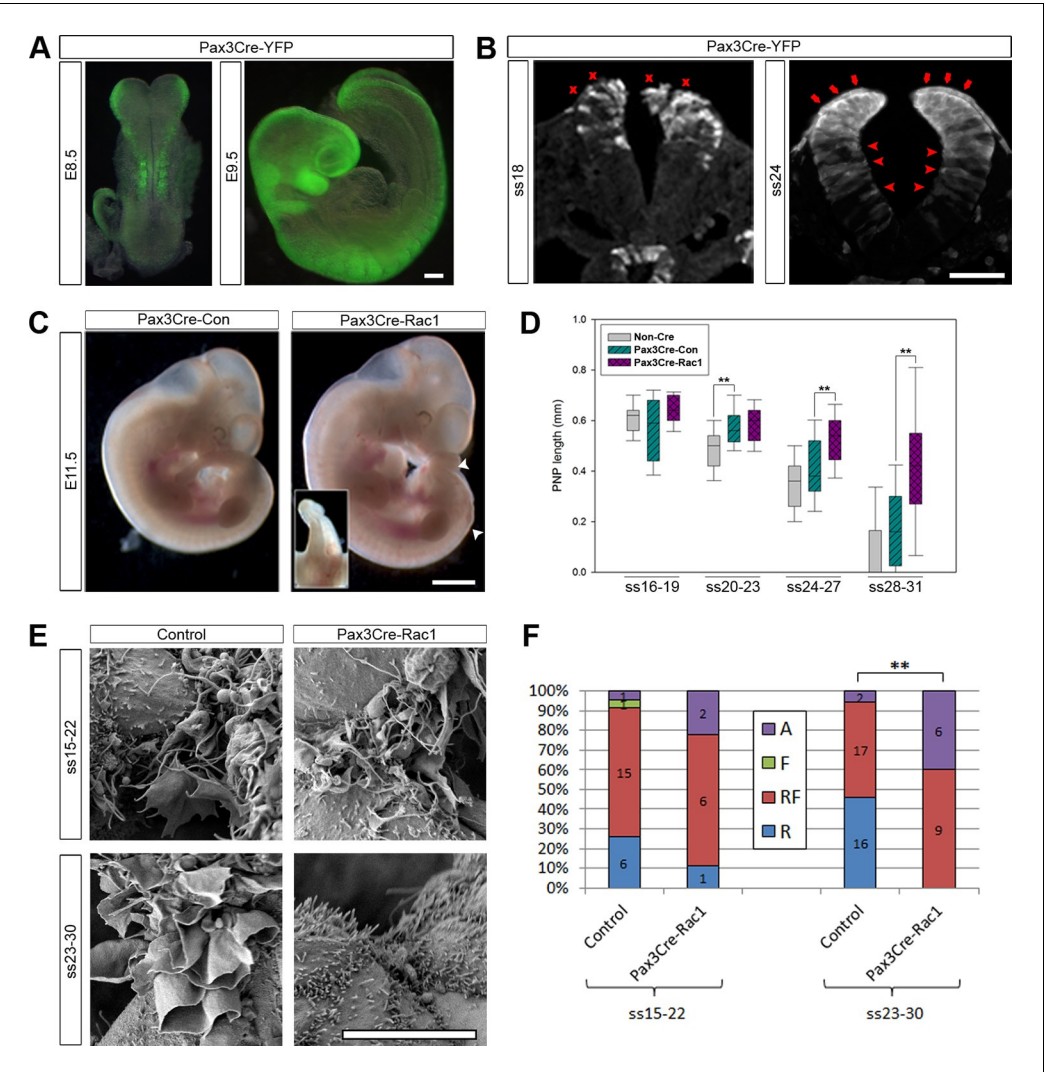

**Figure 3.** Pax3Cre-Rac1 mutants display late failure of PNP closure, with absence of ruffles. (**A, B**) Pax3Cre-driven recombination in the dorsal neural folds and neural tube, detected from E8.5 by direct YFP-reporter expression (**A**), and by immunofluorescence in transverse sections of the closing neural tube at E9.5 (**B**). After ss20, recombination is also detected in the dorsal SE (red arrows), but not at earlier stages (red crosses). Note also recombination in cells of the ventral NE (red arrowheads; see also ***Figure 3—figure supplement 1***). At least three different embryos were analysed for each stage. (**C, D**) Pax3Cre-Rac1 mutants display open spina bifida at E11.5 (**C**, white arrowheads and inset, quantified in ***Table 1***) and delayed PNP closure from ss24-27 onwards (**D**, \*\*p<0.001 – see ***Figure 3—source data 1*** for raw values and statistical details). (**E, F**) SEMs of the PNP fusion point of control embryos show predominantly ruffles and filopodia at ss15-22 and ruffles at ss23-30, whereas Pax3Cre-Rac1 mutants show ruffles and filopodia at ss15-22 and absent protrusions at ss23-30 (**E**, quantified in **F**, p=0.29604 for ss15-22 and \*\*p=0.0002 for ss23-30). A – Absent or incipient protrusions, F – Filopodia only (or predominantly), RF – mixture of Ruffles and Filopodia (or filopodia emanating from ruffles), R – Ruffles only (or predominantly). Scale bars: 100 μm (**A** and **B**), 1 mm (**C**) and 10 μm (**E**).

The following source data and figure supplements are available for figure 3:

**Source data 1.** Source data and statistical analysis for ***Figure 3D***.

**Figure supplement 1.** Pax3Cre drives recombination in a domain of cells that includes the dorsal SE, in addition to dorsal NE.

**Figure supplement 2.** Pax3Cre-Rac1 conditional mutants show tissue-targeted deletion of Rac1.

*Figure 3 continued on next page*

*Figure 3 continued*

**Figure supplement 3.** Pax3Cre-Rac1 mutants show normal bending of the neural plate.

**Figure supplement 4.** Pax3Cre-Rac1 mutants show normal F-actin and adherens junction components distribution.

**Figure supplement 5.** Pax3Cre-Rac1 mutants show defects in neural crest-derived structures.

lacking the *Pax3^Cre* allele, which were either wild-type (floxed) or heterozygous at the *Rac1* locus (denoted Non-Cre). Only 1/80 (1%) of these embryos exhibited spina bifida, not significantly different from the frequency in Pax3Cre-Con embryos (*Table 1*).

A small percentage of Pax3Cre-Rac1 embryos also developed the cranial NTD exencephaly, but at the same low frequency (8%) as was observed in Pax3Cre-Con embryos (*Table 1*). This could reflect the predisposition of Pax3 heterozygotes to exencephaly (*Dempsey and Trasler, 1983*), although this exencephaly frequency was not significantly different from the 1% observed in Non-Cre controls. In any event, the finding of exencephaly in this study is unlikely to be related to the conditional ablation of Rac1.

The open spina bifida lesions in Pax3Cre-Rac1 embryos tended to be small and never extended further anterior than the level of the hindlimb bud (*Figure 3C*). Other embryos displayed a curled tail but no open lesion (not shown) which, in other mouse mutants, can result from delayed spinal neural tube closure (*Copp, 1985*). To assess closure directly, we measured the length of the posterior neuropore (PNP), the region of open spinal neural folds, in embryos between ss16 and ss31. PNP length diminished progressively in control embryos as neurulation proceeded along the spinal region (*Figure 3D*). Comparing Pax3Cre-Rac1 and Pax3Cre-Con embryos, before ss23 there was no detectable difference in PNP length but, from ss24 onwards, the PNP lengths of Pax3Cre-Rac1 embryos were significantly greater than those of Pax3Cre-Con embryos (*Figure 3D*), consistent with a late failure of PNP closure and consequent relatively mild spinal neurulation defects.

At ss20-23, we also detected a significant difference in PNP length between Non-Cre and Pax3-Cre-Con embryos (*Figure 3D*). This is consistent with a delay in PNP closure in *Pax3* heterozygotes (*Auerbach, 1954*; *Dempsey and Trasler, 1983*) and the small percentage of spina bifida observed in these embryos (*Table 1*).

## Rac1 is required for the formation of ruffles during late stages of neural tube closure

To investigate whether the failure of PNP closure in Pax3Cre-Rac1 embryos was accompanied by a defect in the formation of protrusions at the fold apposition point, we analysed this region by SEM at mid (ss15-22) and late (ss23-30) spinal neurulation. Comparing the types of protrusions formed by Non-Cre and Pax3Cre-Con embryos did not reveal a difference between the two groups (p=0.44). Nonetheless, because Pax3 is known to regulate the cytoskeleton in osteogenic cells, and Rac1 activity is required for this function (*Wiggan et al., 2002*; *Wiggan and Hamel, 2002*; *Wiggan et al., 2006*), we also investigated protrusions in the *Pax3* mutant mouse *Sp^2H*, which contains a 32bp deletion in the *Pax3* gene (*Epstein et al., 1991*). No differences were observed between the protrusions of wild-type, heterozygous and *Sp^2H* mutant embryos (data available in doi:10.5061/dryad.rm660). We conclude that *Pax3* heterozygosity does not affect protrusion formation; from here onwards, the protrusion analysis utilised a single category of Controls, comprising pooled Non-Cre and Pax3Cre-Con embryos.

Comparing Pax3Cre-Rac1 and control embryos, there was no significant difference between the proportions of protrusion types formed at ss15-22, whereas at ss23-30 these proportions differed significantly between the two groups. In contrast to control embryos at these late neurulation stages, Pax3Cre-Rac1 embryos never formed ruffles only: they either exhibited both ruffles and filopodia (60% of cases), or had absent or incipient protrusions (40%; *Figure 3F*). These observations are consistent with the failure or delay in PNP closure seen in Pax3Cre-Rac1 embryos resulting from a defect in cell protrusive activity at late neurulation stages. In particular, Pax3Cre-Rac1 embryos are impaired in the formation of membrane ruffles devoid of filopodia.

**Table 1.** Conditional genetic analysis of the roles of Rac1 and Cdc42.

| Cross | Genotype | Abbreviated genotype | Embryonic Day | Total | Exencephaly | Spina bifida and/or curly tail |
|---|---|---|---|---|---|---|
| $Pax3^{Cre/+};Rac1^{f/+}$ X $Rac1^{f/-}$ | | | | Total | Exencephaly | Spina bifida and/or curly tail |
| | $Pax3^{+/+};Rac1^{f/f,\ f/+.\ f/-\ or\ +/-}$ | Non-Cre | 10.5 – 13.5 | 80 | 1 (1%) | 1 (1%) |
| | $Pax3^{Cre/+};Rac1^{f/+\ or\ +/-}$ | Pax3Cre-Con | 10.5 – 13.5 | 39 | 3 (8%) | 2 (5%) |
| | $Pax3^{Cre/+};Rac1^{f/f\ or\ f/-}$ | Pax3Cre-Rac1 | 10.5 – 13.5 | 49 | 4 (8%) | 37 (76%)[**] |
| | | | | Total | Dead or dying | Split face |
| | $Pax3^{+/+};Rac1^{f/f,\ f/+.\ f/-\ or\ +/-}$ | Non-Cre | 13.5 | 43 | 0 | 0 |
| | $Pax3^{Cre/+};Rac1^{f/+\ or\ +/-}$ | Pax3Cre-Con | 13.5 | 17 | 0 | 0 |
| | $Pax3^{Cre/+};Rac1^{f/f\ or\ f/-}$ | Pax3Cre-Rac1 | 13.5 | 21 | 18 (86%)[**] | 21 (100%)[**] |
| $Pax3^{Cre/+};Cdc42^{f/+}$ X $Cdc42^{f/-}$ | | | | Total | Exencephaly | Spina bifida and/or curly tail |
| | $Pax3^{+/+};Cdc42^{f/f,\ f/+.\ f/-\ or\ +/-}$ | Non-Cre | 10.5 – 13.5 | 52 | 0 | 0 |
| | $Pax3^{Cre/+};Cdc42^{f/+\ or\ +/-}$ | Pax3Cre-Con | 10.5 – 13.5 | 17 | 0 | 0 |
| | $Pax3^{Cre/+};Cdc42^{f/f\ or\ f/-}$ | Pax3Cre-Cdc42 | 10.5 – 13.5 | 23 | 0 | 0 |
| | | | | Total | Dead or dying | Split face |
| | $Pax3^{+/+};Cdc42^{f/f,\ f/+.\ f/-\ or\ +/-}$ | Non-Cre | 13.5 | 26 | 1 (4%) | 0 |
| | $Pax3^{Cre/+};Cdc42^{f/+\ or\ +/-}$ | Pax3Cre-Con | 13.5 | 6 | 0 | 0 |
| | $Pax3^{Cre/+};Cdc42^{f/f\ or\ f/-}$ | Pax3Cre-Cdc42 | 13.5 | 11 | 10 (91%)[**] | 11 (100%)[**] |
| $Grhl3^{Cre/+};Rac1^{f/+}$ X $Rac1^{f/f\ or\ f/-}$ | | | | Total | Exencephaly | Spina bifida and/or curly tail |
| | $Grhl3^{+/+};Rac1^{f/f,\ f/+.\ f/-\ or\ +/-}$ | Non-Cre | 10.5 – 13.5 | 141 | 0 | 1 (<1%) |
| | $Grhl3^{Cre/+};Rac1^{f/+\ or\ +/-}$ | Grhl3Cre-Con | 10.5 – 13.5 | 73 | 1 (1%) | 0 |
| | $Grhl3^{Cre/+};Rac1^{f/f\ or\ f/-}$ | Grhl3Cre-Rac1 | 10.5 – 13.5 | 44 | 11 (25%)[**] | 39 (89%)[**] |
| | | | | Total | Unattached allantois | |
| | $Grhl3^{+/+};Rac1^{f/f,\ f/+.\ f/-\ or\ +/-}$ | Non-Cre | 9.5 | 134 | 1 (<1%) | |
| | $Grhl3^{Cre/+};Rac1^{f/+\ or\ +/-}$ | Grhl3Cre-Con | 9.5 | 86 | 0 | |
| | $Grhl3^{Cre/+};Rac1^{f/f\ or\ f/-}$ | Grhl3Cre-Rac1 | 9.5 | 69 | 21 (30%)[**] | |
| $Grhl3^{Cre/+};Cdc42^{f/+}$ X $Cdc42^{f/f}$ | | | | Total | Dead or underdeveloped | |
| | $Grhl3^{+/+};Cdc42^{f/+}$ | Non-Cre | 9.5 – 10.5 | 16 | 0 | |
| | $Grhl3^{Cre/+};Cdc42^{f/f\ or\ f/-}$ | Grhl3Cre-Cdc42 | 9.5 – 10.5 | 12 | 12 (100%)[**] | |
| $Nkx1-2^{CreERT2/+};Rac1^{f/+}$ X $Rac1^{f/-}$ | | | | Total | Exencephaly | Spina bifida and/or curly tail |
| | $Nkx1-2^{+/+};Rac1^{f/f,\ f/+.\ f/-\ or\ +/-}$ | Non-Cre | 10.5 – 13.5 | 51 | 0 | 0 |
| | $Nkx1-2^{CreERT2/+};Rac1^{f/+\ or\ +/-}$ | Nkx1-2Cre-Con | 10.5 – 13.5 | 16 | 0 | 0 |
| | $Nkx1-2^{CreERT2/+};Rac1^{f/f\ or\ f/-}$ | Nkx1-2Cre-Rac1 | 10.5 – 13.5 | 17 | 0 | 0 |

[**]p<0.001 when compared to either Non-Cre or DriverCre-Con.

While it seemed most likely that the neural tube closure defect in Pax3Cre-Rac1 resulted from faulty formation of membrane ruffles, we considered whether conditional deletion of Rac1 might also lead to an earlier defect in neural tube closure, such as NE bending. Failure of dorsolateral bending can cause spina bifida in mice (*Ybot-Gonzalez et al., 2007a*). Transverse sections through the PNP of these embryos at E9.5 confirmed the presence of normal-appearing hinge points at both midline and dorsolateral positions (*Figure 3—figure supplement 3*), arguing against a mechanism of spina bifida in Pax3Cre-Rac1 embryos based on NE bending defects. Nevertheless, because Rac1

is required for the formation of adherens junctions (*Ehrlich et al., 2002*; *Kovacs et al., 2002*; *Yamada and Nelson, 2007*), we analysed the localisation of F-actin and β-catenin in the NE of Pax3-Cre-Rac1 mutants, as well as E-cadherin in the SE. The distribution of these proteins was found to be closely similar in control and mutant embryos, as well as in both the targeted and non-targeted regions of the NE in mutant embryos (*Figure 3—figure supplement 4*), thus ruling out an effect of Rac1 knock-out on epithelial stability, which could impair neural tube closure.

## Failure of PNP closure does not lead to defects in protrusive activity

To address the question of whether the protrusion defects observed in Pax3Cre-Rac1 embryos are indeed a cause of failure of PNP closure, rather than a consequence of that failure, we chose to analyse the cell protrusive activity in a different mouse mutant with spina bifida. The *curly tail (ct)* mutant is homozygous for a hypomorphic allele of the transcription factor *grainyhead-like-3 (Grhl3)* and exhibits spina bifida with 15–20% penetrance (*Gustavsson et al., 2007*; *Ting et al., 2003*; *van Straaten and Copp, 2001*). The size of the spina bifida lesions and the timing of failure of PNP closure in *ct/ct* embryos are similar to those in Pax3Cre-Rac1 embryos. Furthermore, the PNP defect in *ct/ct* embryos is known to be caused by a defect of cell proliferation in the hindgut, leading to excessive curvature of the body axis (*Brook et al., 1991*; *Copp et al., 1988*), and hence is unlikely to be related to protrusion and fusion events at the neural fold tips.

We collected *ct/ct* embryos at ss24-30 and measured their PNP lengths (*Figure 4A* and *Figure 4—figure supplement 1*). For analysis of protrusions, embryos were grouped into two categories: those with PNP lengths above 0.6 mm (large PNP), which are destined to develop spina bifida (*Copp, 1985*) and those with PNP lengths below 0.4 mm (small PNP) that undergo normal PNP closure. Embryos with intermediate sized PNPs were not included in the analysis (*Figure 4—figure supplement 1*). The relative proportions of the different types of protrusions did not differ significantly between embryos with large and small PNPs (*Figure 4B,C*), showing that delay in PNP closure does not cause defects in protrusive activity of cells at the dorsal fusion point.

## Cdc42 is not required in the Pax3 lineage for neural tube closure or protrusion formation

The requirement for Rac1 solely at later stages of spinal neural tube closure raises two possibilities. Rac1 might be required for the formation of ruffles that are devoid of filopodia, but not for the combined appearance of ruffles and filopodia. These ruffles without filopodia might be needed only for late spinal NT closure. Alternatively, it is possible that Rac1 is required for neural fold protrusions along the whole spinal axis throughout all NT closure stages, but these protrusions arise on SE cells, and so Rac1-dependency is seen only at late neurulation stages, when the Pax3 lineage becomes targeted in the SE as well as the NE (*Figure 3B* and *Figure 3—figure supplement 1*). To address the first hypothesis, and to test a possible candidate for the formation of filopodia, we generated conditional Cdc42 mutants in the Pax3 cell lineage (Pax3Cre-Cdc42). None of the Pax3Cre-Cdc42 embryos showed NTDs (*Table 1*). Moreover, Pax3Cre-Cdc42 embryos collected during neurulation between ss16 and ss27 resembled control litter mates in PNP length (*Figure 5A*). The proportions of protrusion types at mid (ss15-22) and late (ss23-30) neurulation did not differ between Pax3Cre-Cdc42 and control embryos (*Figure 5B,C*). These results suggest that the protrusions seen at earlier neurulation stages are either regulated independently of both Rac1 and Cdc42, or may emanate from SE rather than NE cells.

## Rac1 and Cdc42 are required in the Pax3 lineage for development of neural crest-derived structures

In view of the different effects on neural fold protrusions that we observed when deleting *Rac1* and *Cdc42* in the Pax3 cell lineage, it was important to confirm that Pax3Cre-Rac1 and Pax3Cre-Cdc42 embryos both developed expected phenotypes later in development. These small GTPases are required in the neural crest for proper development of structures including craniofacial primordia, dorsal root ganglia, and the cardiac outflow tract septum (*Fuchs et al., 2009*; *Thomas et al., 2010*). The Pax3 lineage encompasses the neural crest population (*Engleka et al., 2005*), and therefore defects in neural crest derivatives were expected in both Pax3Cre-Rac1 and Pax3Cre-Cdc42 embryos. Embryos of both genotypes were dead by E13.5 (*Table 1*), as expected in cases of

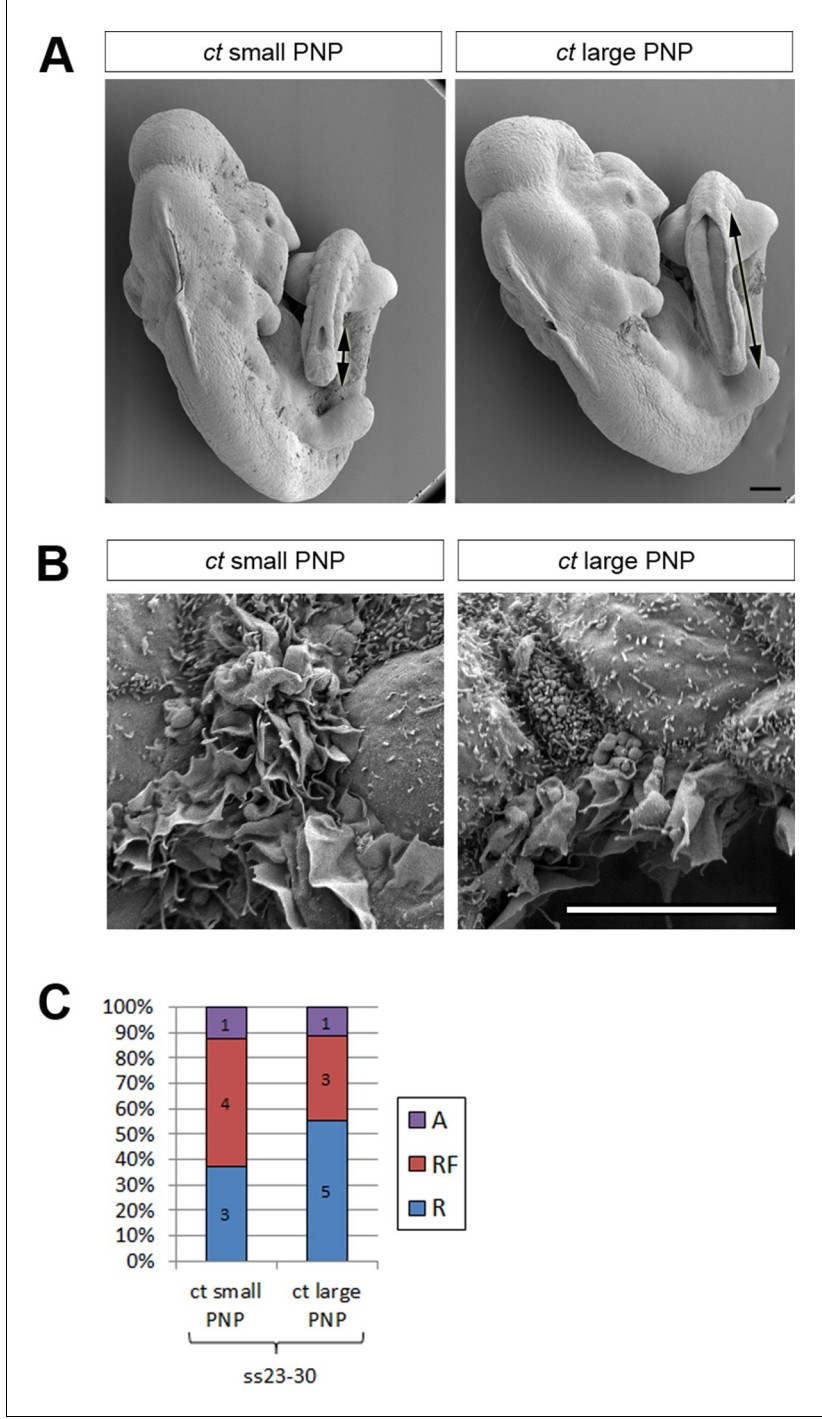

**Figure 4.** Failure of PNP closure does not cause defective protrusive activity. (**A**) SEMs of E9.5 *curly tail* embryos showing examples of small and large PNPs (double arrows). (**B, C**) SEMs of the PNP fusion point of *curly tail* embryos show either membrane ruffles or ruffles and filopodia at ss23-30 (**B**, quantified in **C**). There is no difference in protrusion type or frequency between embryos with small and large PNPs (p=0.71782). Definition of protrusion types as in *Figure 3*. Scale bars: 100 μm (**A**) and 10 μm (**B**).

The following figure supplement is available for figure 4:

**Figure supplement 1.** Size range of PNPs of *curly tail* embryos collected at E9.5 and their grouping according to PNP length.

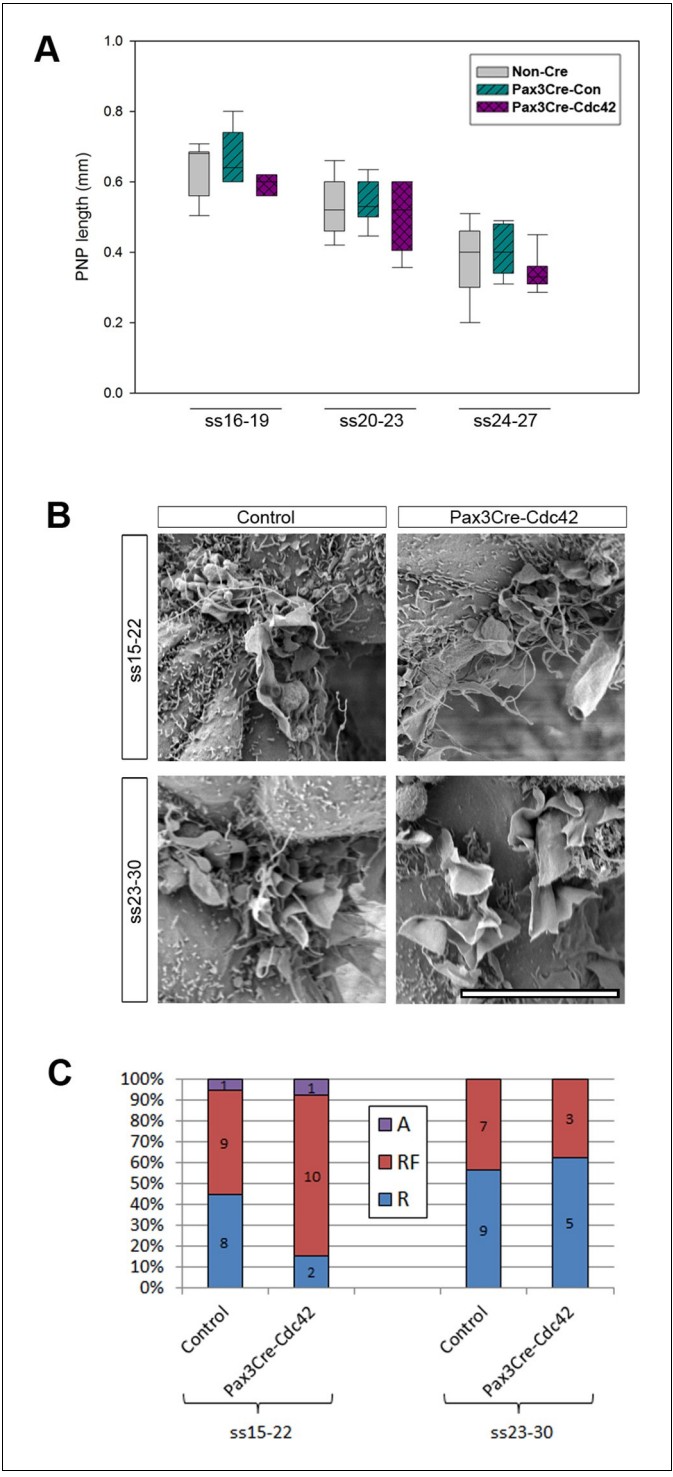

**Figure 5.** Pax3Cre-Cdc42 mutants do not show defects in neural tube closure or protrusive activity. (**A**) Pax3Cre-Cdc42 mutants show a normal rate of PNP closure from ss16-27 (see *Figure 5—source data 1* for raw values and statistical details). (**B**, **C**) SEMs of the PNP fusion point of control and Pax3Cre-Cdc42 embryos show no difference in the types of protrusions formed at ss15-22 and ss23-30 (**B**, quantified in **C** –p=0.0.1533 and p=0.36722 for ss15-22 and ss23-30, respectively). This is consistent with the lack of spina bifida seen in Pax3Cre-Cdc42 mutants (see *Table 1*). Definition of protrusion types as in *Figure 3*. Scale bar: 10 μm (**B**).

The following source data and figure supplement are available for figure 5:

*Figure 5 continued on next page*

*Figure 5 continued*

**Source data 1.** Source data and statistical analysis for *Figure 5A*.
**Figure supplement 1.** Pax3Cre-dc42 mutants show defects in neural crest-derived structures.

defective outflow tract septation (*Conway et al., 1997*). Moreover, both displayed a split face phenotype from E11.5 onwards, consistent with a defect in neural crest-related craniofacial development (*Table 1*, *Figure 3—figure supplement 5A* and *Figure 5—figure supplement 1A*). Transverse sections at the heart level of E12.5 embryos confirmed the absence of an outflow tract septum in these embryos, as well as dorsal root ganglia that were severely reduced in size (*Figure 3—figure supplement 5B* and *Figure 5—figure supplement 1B*). Transverse sections further down the spinal cord at the hindlimb level, where spina bifida occurs in Pax3Cre-Rac1 embryos, also showed severely reduced dorsal root ganglia (*Figure 3—figure supplement 5C* and *Figure 5—figure supplement 1C*), thus confirming effective knockdown of Rac1 and Cdc42 protein throughout the spinal cord. Additionally, Pax3Cre-Cdc42 embryos displayed an apparent disorganisation of the dorsal neural tube, including occlusion of the lumen (*Figure 5—figure supplement 1C*), consistent with a role described for Cdc42 in the polarity and organisation of the layers in the mouse brain (*Cappello et al., 2006*).

## Rac1 is required in the SE, but not in the NE, for neural tube closure and protrusion formation

To test the hypothesis that the protrusions observed at the neural fold fusion point emanate from SE cells, we generated a conditional knock-out for *Rac1* in this tissue by crossing *Rac1* floxed mice with mice expressing Cre under the control of *Grhl3* (see Materials and Methods). *Grhl3* is expressed predominantly in the SE during early neurulation (*Gustavsson et al., 2007*; *Ting et al., 2003*), and Grhl3Cre has been used as an early SE driver (*Camerer et al., 2010*; *Massarwa and Niswander, 2013*; *Ray and Niswander, 2016*). Grhl3Cre-Rac1 mice were generated previously and found to have NTDs, including highly penetrant spina bifida, but cell protrusion analysis was not performed (*Camerer et al., 2010*).

We confirmed that Grhl3Cre drives recombination throughout the SE from E8.5, and continuing at all stages of neurulation. We also detected recombination in a small proportion of scattered neuroepithelial cells (*Figure 6A,B*), consistent with reported expression of *Grhl3* in the neuroepithelium at E9 (*Gustavsson et al., 2007*). *Rac1* ablation in Grhl3Cre-Rac1 embryos was confirmed by mRNA in situ hybridisation (*Figure 6—figure supplement 1*). A proportion (30%) of Grhl3Cre-Rac1 embryos displayed failure of chorioallantoic fusion (*Table 1*; data not shown), a phenotype not previously described in these mice, which was accompanied by defects of growth and axial rotation that can result from such failure (*Morin-Kensicki et al., 2006*; *Saunders et al., 2004*; *Stumpo et al., 2004*). These embryos were excluded from further analysis. Most of the remaining Grhl3Cre-Rac1 embryos had cranial and/or spinal NTDs: 25% exhibited exencephaly and 89% had spina bifida (*Table 1*). The size of the spina bifida lesions in these embryos was larger than that observed in Pax3Cre-Rac1 embryos, usually starting rostral to the hindlimb bud (*Figure 6C*), which suggested an earlier failure of PNP closure. To examine this possibility, we measured the PNP of embryos at ss16-27 and detected a significant delay in PNP closure in Grhl3Cre-Rac1 embryos from ss20-23, compared with Grhl3Cre-Con and NonCre-Con embryos (*Figure 6D*). This confirmed that Grhl3Cre-Rac1 embryos fail to close their PNP earlier than Pax3Cre-Rac1 mutants.

Analysis of transverse sections through the PNP at E9.5 revealed normal-appearing dorsolateral hinge points in Grhl3Cre-Rac1 embryos, thus arguing against a neural plate bending defect as the underlying cause for the spina bifida phenotype (*Figure 6—figure supplement 2*).

We next analysed the protrusive activity at the PNP fusion point in Grhl3Cre-Rac1 embryos. Despite only reaching statistically significant difference from controls at ss23-30, Grhl3Cre-Rac1 embryos never showed membrane ruffles at any stage analysed, and also showed an increased occurrence of filopodia without associated ruffles (*Figure 6E,F*). Furthermore, the density and length of filopodia in mutants appeared increased (*Figure 4E*), and this was confirmed by measuring the number of filopodia present around the point of fusion, as well as their length (see 'Materials and

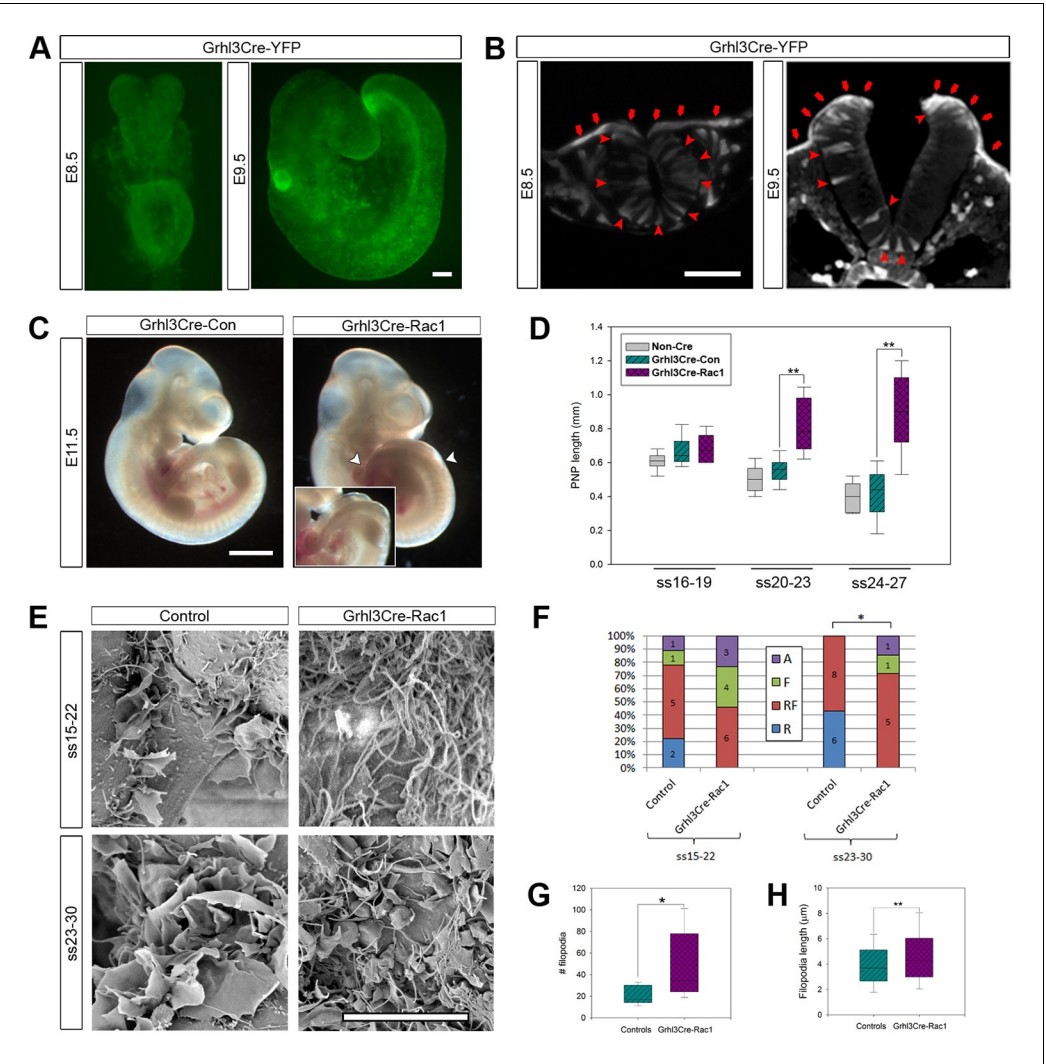

**Figure 6.** Grhl3Cre-Rac1 mutants show failure of PNP closure from ss20-23, accompanied by abnormal protrusive activity. (**A, B**) Grhl3Cre-driven recombination in the SE is detected from E8.5 by direct YFP-reporter expression (**A**), as well as by immunofluorescence in transverse sections of the closing neural tube (**B**, red arrows). Note additional recombination in randomly scattered cells in the NE (**B**, red arrowheads) and other tissues. At least three different embryos were analysed for each stage. (**C, D**) Grhl3Cre-Rac1 mutants display open spina bifida at E11.5 (**C**, between white arrowheads and inset, quantified in *Table 1*) and delayed PNP closure from ss20-23 (**D**, **p<0.001 – see *Figure 6—source data 1* for raw values and statistical details). (**E–H**) SEMs of the PNP fusion point of Grhl3Cre-Rac1 embryos show protrusive activity that differs from control embryos. Filopodia without ruffles are observed in Grhl3Cre-Rac1 embryos, especially at ss15-22, and membrane ruffles without filopodia are never detected (**E**, quantified in **F**, p=0.27024 for ss15-22 and *p=0.02735 for ss23-30). Definition of protrusion types as in *Figure 3*. In the cases where filopodia where present (RF and F categories), these were present in higher number (**G**) and were longer (**H**) in Grhl3Cre-Rac1 embryos when compared to Controls (*p<0.05, **p<0.001, see *Figure 6—source data 2* for raw values and statistical details). Scale bars: 100 μm (**A** and **B**), 1 mm (**C**) and 10 μm (**E**).

The following source data and figure supplements are available for figure 6:

**Source data 1.** Source data and statistical analysis for *Figure 6D*.

**Source data 2.** Source data and statistical analysis for *Figure 6G,H*.

**Figure supplement 1.** Grhl3Cre-Rac1 conditional mutants show tissue-targeted deletion of Rac1.

*Figure 6 continued on next page*

*Figure 6 continued*

**Figure supplement 2.** Grhl3Cre-Rac1 mutants show normal bending of the neural plate.

methods' for details) (*Figure 6G,H*). These results indicate that, like in late neurulation, cell protrusions emanate from SE cells during earlier neurulation and are at least partly regulated by Rac1.

## Rac1 is not required in the NE for successful neural tube closure

The analysis of Pax3Cre-Rac1 and Grhl3Cre-Rac1 embryos suggested that Rac1 is required in the SE for the regulation of membrane protrusions: the implication being that the protrusions emanate from SE not NE cells. However, since Cre recombination also occurred in dorsal NE cells in Pax3Cre lines, and in scattered NE cells in Grhl3Cre lines, we could not rule out a role of Rac1 in the NE. To resolve this issue, we performed a further experiment to test specifically whether Rac1 function in NE cells may mediate cell protrusions and neural tube closure. We used Nkx1-2Cre-ERT2 (*Rodrigo Albors et al., 2016*) to generate Nkx1-2Cre-Rac1 embryos in which *Rac1* was conditionally inactivated solely in NE cells. *Nkx1-2,* also known as *Sax1,* is a homeobox gene expressed in the posterior neural tube (*Schubert et al., 1995*). In situ hybridisation confirmed that *Nkx1-2* is expressed solely in the closing PNP of E9.5 embryos (*Figure 7A*), with transcripts detectable only in the NE (*Figure 7B*). Using the ROSA26-EYFP reporter, we confirmed that Nkx1-2Cre drives recombination in the NE of the PNP and previously closed neural tube (*Figure 7C*), whereas there is no recombination in the SE (*Figure 7D*). Nkx1-2Cre-Rac1 embryos developed entirely normally, and did not display NTDs (*Table 1*). Moreover, they formed normal ruffles typical of late neurulation (*Figure 7E,F*). This experiment demonstrates unequivocally that Rac1-dependent cellular protrusions do not emanate from the NE during spinal closure.

## Cdc42 is required for the formation of filopodia during early neurulation

Our finding of increased density and length of filopodia in Grhl3Cre-Rac1 mutants (*Figure 6E,F*) suggested that, rather than being required for the formation of filopodia, Rac1 may in fact be needed to suppress them and maintain a balance between the formation of the different kinds of protrusions. To test the role of Cdc42 in the formation of filopodia in early neurulation, we conditionally ablated *Cdc42* predominantly in the SE lineage, using Grhl3Cre. The *Grhl3* and *Cdc42* genes are located less than 2 Mb apart on mouse chromosome 4 (www.ensembl.org), and therefore have less than 2% chance of recombination. In initial crosses, we obtained a rare recombinant in which the *Grhl3$^{Cre}$* and *Cdc42$^f$* alleles were in coupling, allowing the necessary *Grhl3$^{Cre/+}$; Cdc42$^{f/+}$* x *Grhl3$^{+/+}$; Cdc42$^{f/f}$* mating to be performed, but precluding the generation of *Grhl3$^{Cre/+}$; Cdc42$^{f/+}$* controls for this cross (*Table 1*; see 'Materials and methods' for details).

Remarkably, Grhl3Cre-Cdc42 embryos were dead by E10.5 (*Table 1*), and at E9.5 were already severely growth retarded, having failed to undergo axial rotation (*Figure 8A*). These embryos underwent normal chorioallantoic fusion (not shown), unlike some Grhl3Cre-Rac1 mutants. Due to their early lethality, we analysed the protrusive activity of Grhl3Cre-Cdc42 embryos at E8.5. The initial closure event (Closure 1) occurred normally at ss6-7, and Grhl3Cre-Cdc42 embryos at ss6-10 were indistinguishable from control littermates (*Figure 8B*). At the spinal closure point, cellular protrusions differed significantly from control embryos: they were predominantly ruffles, in sharp contrast with the predominance of filopodia seen in controls at these early neurulation stages (*Figure 8C,D*). This finding suggests that Cdc42 is required for filopodia formation during early spinal closure.

## Protrusive cells have SE-like cell morphology

Our genetic experiments indicated that the cell protrusions required for neural tube closure emanate from SE cells. To examine this question by a different methodology, we performed serial block-face SEM of the PNP fusion point (*Figure 9A–C*). This technique allows high-resolution imaging of cells coupled with the ability to perform three-dimensional reconstructions which allow analysis of the entire cell shape, thus combining the advantages of TEM and SEM (*Hughes et al., 2014*). In the case of the closing spinal neural folds, we were able to identify the membrane protrusions and their cell of origin in each section, and thus trace them in all the sections to finally obtain a reconstruction

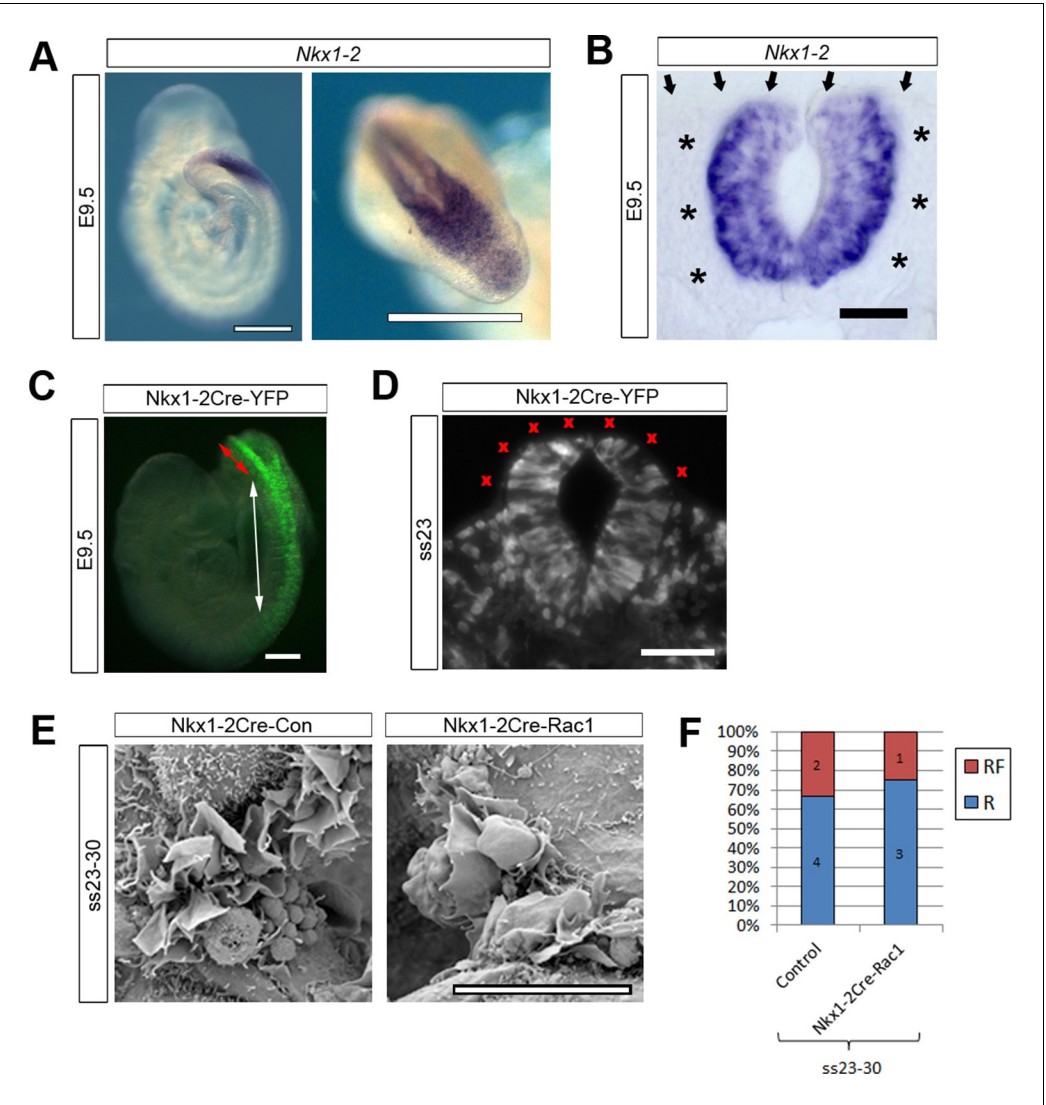

**Figure 7.** Nkx1-2Cre is expressed in NE and Nkx1-2Cre-Rac1 mutants display normal cellular protrusions. (A, B) In situ hybridisation for *Nkx1-2* in whole mount E9.5 embryos. (A) Left: lateral view; right: dorsal view of the PNP. *Nkx1-2* transcripts are confined to the neural plate and very recently closed neural tube. A transverse vibratome section at the level of the closing PNP (B) reveals *Nkx1-2* expression solely in the NE, and not in adjacent mesoderm (asterisks) nor overlying SE (arrows). (C, D) Nkx1-2Cre-driven recombination in the closing neural tube detected by immuno-fluorescence of YFP-reporter expression at E9.5. Note the presence of YFP in the NE of the closing PNP (red double-arrow) and previously-closed neural tube (white double-arrow) (C), but not in the SE lateral to the NE. A transverse section through the closing neural tube at E9.5 (D) shows the complete absence of YFP from the SE (red crosses; 10 embryos analysed). (E, F) Nkx1-2Cre Rac1 mutants have no neurulation defects (see *Table 1*) and SEMs of their PNP fusion point at ss24-30 show predominantly membrane ruffles, similar to control embryos (E, quantified in F, p=0.75). Definition of protrusion types as in *Figure 3*. Scale bars: 500 μm (A), 50 μm (B), 100 μm (C, D), and 10 μm (E).

of the entire protrusive cell shape (*Figure 9* and *Video 1*). We also traced non-protrusive SE and NE cells for comparison.

SE cells have a simple squamous epithelial morphology, with a short apico-basal dimension (yellow cell, *Figure 9D,E,G*, and *Video 1*), whereas NE cells have the typical shapes of a pseudo-stratified epithelium, with a long apico-basal axis, and wedge, spindle or inverted wedge morphology, depending on nuclear position (dark blue and cyan cells, *Figure 9D,E,G*, and *Video 1*), as described previously in the neural plate (*Schoenwolf, 1985*; *Smith et al., 1994*). We confirmed that, in each

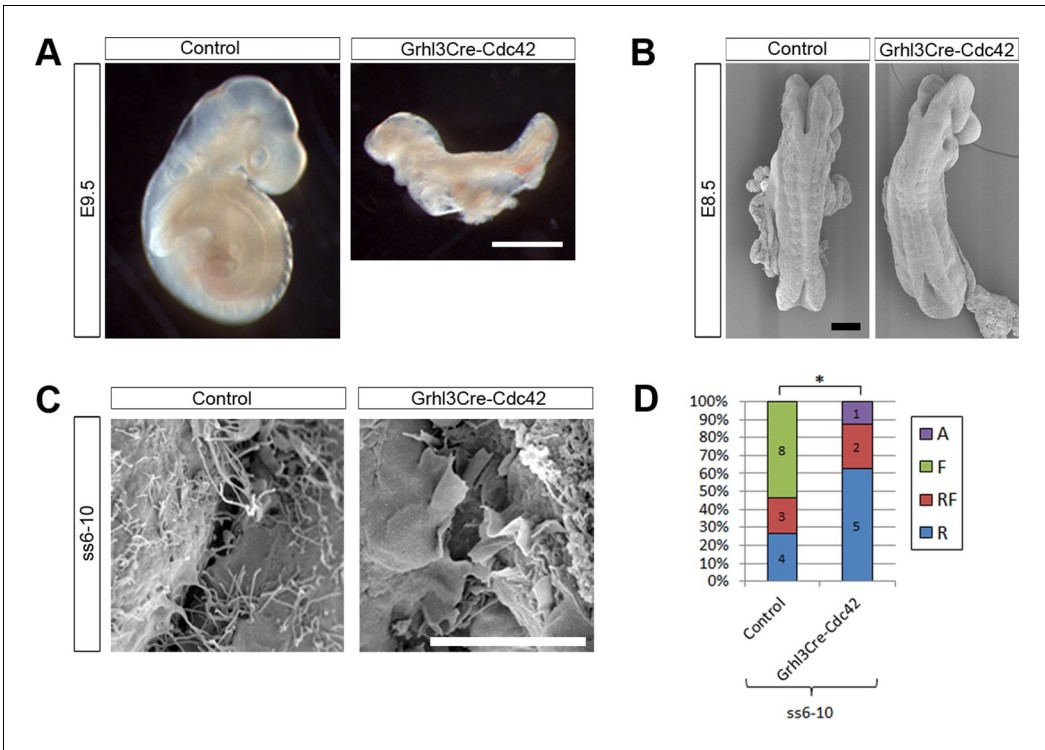

**Figure 8.** Grhl3Cre-Cdc42 embryos show altered protrusive activity. (A) Grhl3Cre-Cdc42 mutants have an embryonic lethal phenotype, with E9.5 embryos displaying reduced size and failure of axial rotation (quantified in *Table 1*). (B) SEMs of E8.5 embryos with fewer than 11 somites. At this stage, Grhl3Cre-Cdc42 mutants are indistinguishable in overall morphology from control littermates. (C, D) SEMs of the PNP fusion point of Grhl3Cre-Cdc42 mutants at ss6-10 show a predominance of membrane ruffles, in contrast to the filopodia seen in control embryos at this stage (C, quantified in D, *p<0.05). Definition of protrusion types as in *Figure 3*. Scale bars: 1 mm (A), 100 μm (B) and 10 μm (C).

transverse section, membrane protrusions emanated from only a single pair of cells bilaterally, one on each neural fold tip. Moreover, these cells were positioned precisely at the junction between the SE and NE. Their cell bodies were flat in the apico-basal direction and elongated in the plane of the tissue, closely similar to SE cells, except for the elaborate membrane protrusive activity on their free ends (red and green cells, *Figure 9D–F,H*, and *Video 1*). We conclude that the protrusive cells are most likely of SE origin, although we cannot rule out the hypothesis that they might be highly modified NE cells.

## Discussion

The presence of cell protrusions at the edges of neural folds in rodent embryos was first described several decades ago, but their role in neurulation has remained unknown. The present work is, to our knowledge, the first to provide evidence that these protrusions are required for successful neural tube closure. Moreover, we show that as spinal neurulation progresses these cell protrusions change, both qualitatively and in their genetic regulation. At the onset of neurulation, the protrusions are predominantly filopodial, requiring Cdc42 function, whereas later stages are characterised by either a mixture of ruffles and filopodia, or by membrane ruffles alone, and these require Rac1. Additionally, using both genetic experiments and morphological analysis, we show that the protrusions originate from SE cells rather than NE cells.

### Roles of cell protrusions in epithelial fusion

Besides neural tube closure, other morphogenetic events involve epithelial tissue fusions accompanied by cellular protrusive activity. For example, during palatal shelf fusion in mice, filopodia are

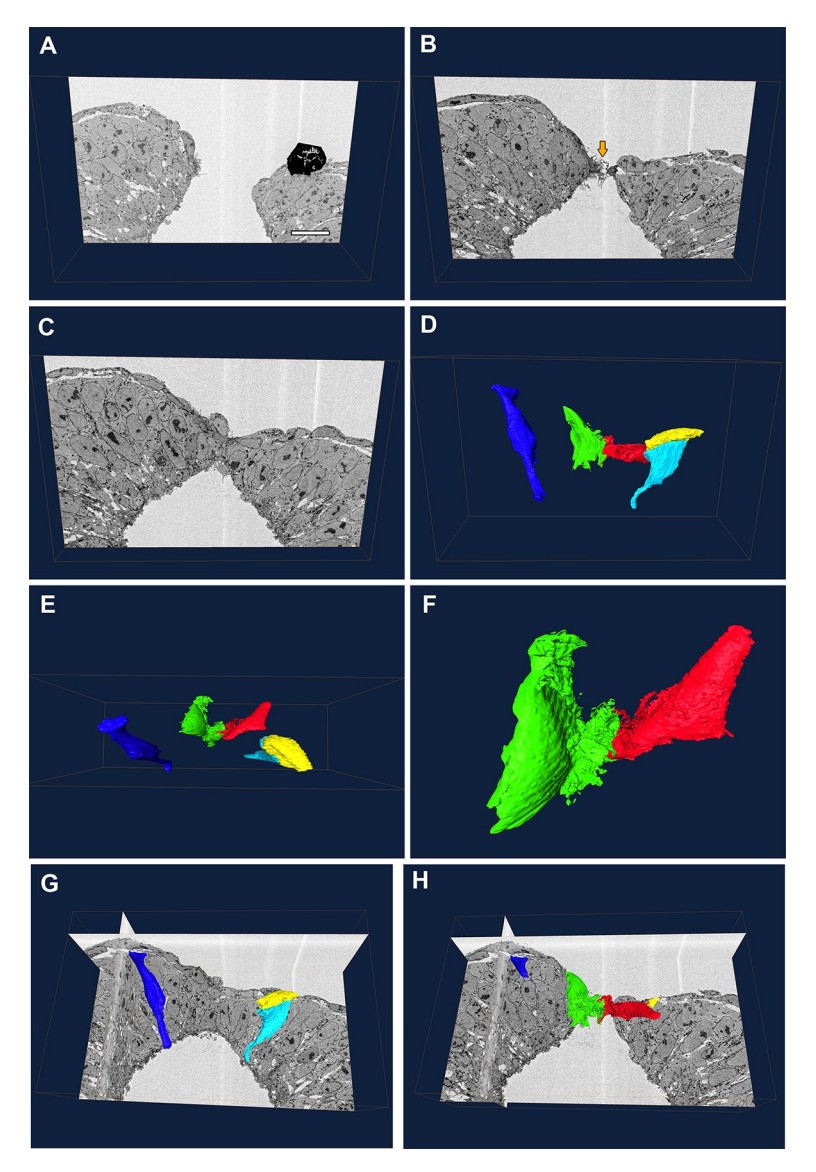

**Figure 9.** Protrusive cells have a SE-like morphology. (**A–F**) Still images from *Video 1*. (**A–C**) A series of transverse section images obtained through serial block-face SEM imaging of the closure point of the PNP at E9.5. Protrusions are visible at the tips of the neural folds (orange arrow in **B**). Black object in (**A**) is an artefact. (**D–F**) Three-dimensional reconstructions of different cell types from the same section-stack. Examples are shown of typical morphologies of pseudostratified NE cells (dark blue: spindle shaped cell; cyan: wedge shaped cell) and of a squamous SE cell (yellow). A single pair of cells are extending protrusions, one from each neural fold (green and red), and these have a squamous-type cell morphology, similar to SE cells. Relative to the sections in **A–C**, the reconstructed cell volumes are shown in the same orientation (**D**), rotated 90° forward (**E**) or rotated 90° forward with zoom (**F**). (**G, H**) Orthoslices from the analysed stack with superimposed three-dimensional reconstructions of the cells described above. Three different embryos were analysed at ss20-26, with similar results. Scale bar: 100 µm (**A**).

present at the medial edge epithelial cells just before fusion, and *TGF-β3* knock-out mice that lack such protrusions display cleft palate (*Taya et al., 1999*). Similarly, during eyelid closure, filopodia extend from the leading edge epithelia and are reduced in number and length in *c-jun* mutants that display defective eyelid closure (*Zenz et al., 2003*). But perhaps the best studied case of cell protrusive activity in epithelial fusion during development is the process of dorsal closure in *Drosophila*. In

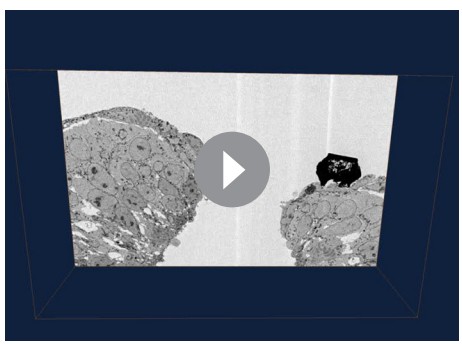

**Video 1.** Serial block-face SEM reveals that protrusive cells have a SE-like morphology. Animation showing a series of transverse images of the spinal neural tube closure point at E9.5, with superimposed three-dimensional reconstructions of NE cells (dark blue and cyan), a SE cell (yellow), and protrusive cells at the tips of the neural folds (green and red). See *Figure 9* legend.

this system, the leading edge cells of the lateral epidermis extend cell protrusions as they advance over the amnioserosa layer. When Rac or Cdc42 functions are perturbed, using dominant-negative proteins or loss-of-function mutations, this leads to defects of lamellipodia or filopodia, respectively, accompanied by failed dorsal closure, as well as misalignment of any segments that manage to close (*Harden et al., 1995*; *1999*; *Jacinto et al., 2000*; *Woolner et al., 2005*; *Hakeda-Suzuki et al., 2002*). This suggests that in *Drosophila,* these protrusions may have a dual role, both in terms of mechanically participating in the closure process, and as exploratory structures that assure proper matching of fusing segments. However, because Rac is also involved in contraction of the underlying amnioserosa cells (*Harden et al., 1995*; *2002*), it is possible that its role in the leading edge cells is mainly exploratory.

Unlike the lateral epidermis leading edge cells in *Drosophila*, which are moving over the amnioserosa, the protrusions that emanate from the mouse neural folds are not crawling on top of other cells or on extracellular matrix; rather, they extend into a fluid-filled space, and therefore could not be exerting any type of traction force to drive fusion. Moreover, in the segmented epidermis of the early *Drosophila* embryo, exact alignment of A-P segments appears vital and is sub-served by the epithelial protrusions (*Millard and Martin, 2008*). In contrast, the mouse developing spinal cord is not overtly segmented and it would appear that the role of neural fold protrusions is more likely to involve cell-cell recognition and/or signalling across the midline, to initiate epithelial fusion and ensure robust closure of the neural tube.

Cell-cell recognition is likely a key step in triggering fusion, and many epithelial fusions involve cell recognition through Eph-ephrin interactions, including palatal shelf development (*Compagni et al., 2003*; *Davy et al., 2004*; *Risley et al., 2009*), optic fissure closure (*Noh et al., 2016*), and neurulation (*Abdul-Aziz et al., 2009*; *Holmberg et al., 2000*). The EphA2 receptor is present on the mouse spinal neural folds just before fusion, and its expression can be detected by TEM on the protrusions themselves (*Abdul-Aziz et al., 2009*), raising the possibility that Eph-ephrin signalling may initiate upon contact between cell protrusions from apposing cells.

Epithelial cell protrusions may also be involved in initiation of de novo cell adhesions. In cultured MDCK cells, E-cadherin accumulation is induced by contacts between Rac1-driven exploratory lamellipodia from opposing cells. The initial contact then spreads, driven by actomyosin contraction, while Rac1 activity and lamellipodial extension cease, and new junctions are formed between the two cells (*Yamada and Nelson, 2007*).

What drives the formation of protrusions in the first place? Recent studies in *Drosophila* suggest that during dorsal closure the epithelial leading edge cells undergo an incomplete epithelial-to-mesenchymal transition, caused by loss of apico-basal polarity (*Bahri et al., 2010*; *Pickering et al., 2013*). Loss of polarity in these cells results in a reduction of PTEN phosphatase, which in turn causes an increase in PIP3 (*Pickering et al., 2013*), a known activator of Rac. As the cells meet in the midline, they switch back to 'full' epithelial character and restore cell-cell adhesion, an event mediated by Pak, an effector of both Rac and Cdc42 (*Bahri et al., 2010*).

## Different requirements for Rac1 and Cdc42 throughout spinal neurulation

Our results with Pax3Cre-Rac1 mutants show a clear requirement for Rac1 and ruffle formation on SE cells during the final stages of primary neurulation, from ss24 onwards. However, Rac1 is required in the SE earlier on, as Grhl3Cre-Rac1 mutants show delayed PNP closure from ss20, and a lack of ruffles alone in these mutants is detected at ss15-22, despite no overall statistical significance. This

argues for a requirement of ruffles from at least mid-neurulation, and perhaps a gradual transition between protrusive types, with a balance of different protrusions needed.

Despite the lack of ruffles, the shift towards excessive filopodia formation in Grhl3Cre-Rac1 mutants was unexpected given that, in cultured fibroblasts, expression of a dominant-negative form of Rac1 leads to inhibition of filopodia (*Johnston et al., 2008*). Fibroblasts genetically deficient for Rac1, on the other hand, are able to spread and move by extending filopodia, possibly through an Arp2/3-independent process (*Steffen et al., 2013*). Filopodial initiation can be driven by either Arp2/3 (branched nucleation) or by formins (unbranched nucleation) (*Yang and Svitkina, 2011*). Moreover, Rac1 can both activate and inhibit the Arp2/3 complex, through either the Scar/WAVE complex or the Arpin protein, respectively (*Dang et al., 2013*). If filopodium formation on the leading edge SE cells occurs through an Arp2/3-dependent process, then in the absence of Rac1 perhaps initiation of filopodial extension can be driven by other activators of the Arp2/3 complex. The occurrence of filopodia initiation would then be enhanced if Rac1 activates an Arp2/3 inhibitor such as Arpin in these cells. On the other hand, if initiation of these filopodia is driven independently of Arp2/3, then the absence of Rac1 might shift the balance from branched to unbranched actin nucleation, resulting in more actin filaments being incorporated into filopodium-forming cross linked bundles.

Our results show that Cdc42 is not needed in mid and late spinal neurulation, as the Pax3Cre-Cdc42 mutants have no defects in PNP closure or protrusive activity. But Cdc42 does play a role in early neurulation, as Grhl3Cre-Cdc42 mutants show a shift towards the extension of ruffles at the expense of filopodia. Whether or not this would impair progression of neural tube closure past ss10 could not be determined, given the early lethality of these embryos. It is possible that, in the absence/reduction of filopodia, membrane ruffles could take over their role and closure would progress. In fact, mutant mice for Ena/VASP proteins (actin regulators involved in filopodium formation) successfully close their PNP despite having cranial neural tube closure defects (*Kwiatkowski et al., 2007*; *Lanier et al., 1999*; *Menzies et al., 2004*), arguing that filopodia are dispensable for spinal neurulation.

## Materials and methods

### Mouse procedures

Animal studies were performed according to the UK Animals (Scientific Procedures) Act 1986 and the Medical Research Council's *Responsibility in the Use of Animals for Medical Research* (July 1993). Non-mutant embryos were from random-bred CD1 mice for standard SEM analysis, and BALB/c for serial block-face imaging SEM. *Curly tail* mice were maintained as a random-bred homozygous colony (*Gustavsson et al., 2007*). Cre-driver lines were *Pax3*^Cre/+ (*Engleka et al., 2005*), *Grhl3*^Cre/+ (*Camerer et al., 2010*) and *Nkx1-2*^CreERT2/+ (*Rodrigo Albors et al., 2016*). Floxed lines were *Rac1*^f/f (*Glogauer et al., 2003*), *Cdc42*^f/f (*Wu et al., 2006*), and *ROSA26-EYFP* (*Srinivas et al., 2001*), all maintained on a C57BL/6 background.

For the generation of conditional mutants, the following general scheme was followed (where 'Driver' refers to either *Pax3, Grhl3* or *Nkx1-2*, and 'GTPase' refers to *Rac1* or *Cdc42*): heterozygous floxed lines were initially crossed with mice carrying the ubiquitously expressed transgene *Actb-Cre* (*Lewandoski and Martin, 1997*) to generate heterozygous *Actb-Cre*^tg; GTPase^+/- mice, which were then back-crossed to GTPase^f/f to generate heterozygous GTPase^f/- (with removal of the *Actb-Cre*). Doubly heterozygous Driver^Cre/+; GTPase^f/+ mice were generated and crossed with GTPase^f/- mice to obtain conditional mutants. This scheme was altered when the *Grhl3*^Cre/+ line was found to drive recombination in the germ line of about 50% of the progeny (not shown), and in that case the crosses were *Grhl3*^Cre/+; GTPase^f or -/+ X GTPase^f/f. For the crosses with *Nkx1-2*^CreERT2/+, Cre activation was induced by intraperitoneal injection of pregnant mothers with a mixture of 20 mg/ml Tamoxifen (Sigma) and 10 mg/ml Progesterone (Sigma), total volume 75 µl, at both E7.5 and E8.5.

Embryos were dissected in DMEM (Invitrogen) containing 10% fetal bovine serum (Sigma) and rinsed in PBS prior to fixation. Yolk sacs were used for embryo genotyping.

## Immunofluorescence

Embryos were fixed for at least 2 hr in 4% paraformaldehyde in PBS, pH 7.4, at 4°C, and dehydrated in a methanol series, except for the embryos stained for F-actin. Immunofluorescence was performed on 12-µm-thick cryosections of gelatine-embedded embryos (7.5% gelatine [Sigma] in 15% sucrose). F-actin was detected using Alexa-Fluor-568–phalloidin (Life Technologies A12380). β-catenin was detected using a rabbit polyclonal antibody (Abcam ab16051). YFP was detected using anti-GFP rabbit polyclonal Alexa488-conjugated antibody (Life Technologies A21311) at 1:1500 dilution (for single-label detection) or anti-GFP chicken polyclonal antibody (Abcam ab13970) at 1:500 dilution (for double-labelling with E-cadherin). E-cadherin was detected using a rabbit monoclonal antibody (Cell Signaling Technology 23E10) at 1:100 dilution. Pax3 was detected using a 1:50 dilution of mouse anti-Pax3 monoclonal antibody concentrate (Developmental Studies Hybridoma Bank, created by the NICHD of the NIH and maintained at The University of Iowa, Department of Biology, Iowa City, IA 52242). For E-cadherin and Pax3, epitopes were unmasked by boiling three times for 3 min in citrate buffer. Secondary antibodies were goat anti-rabbit Alexa568 (A21069), goat anti-rabbit Alexa488 (A11070), goat anti-mouse Alexa568 (A11004), and goat anti-chicken Alexa488 (A11039) (all Life Technologies), all at 1:500 dilution. Images were captured on an Olympus IXZ1 inverted microscope or on an LSM710 confocal system mounted on an Axio Observer Z1 microscope (Carl Zeiss Ltd, UK), and linear adjustments made using Fiji software.

## Whole-mount mRNA in situ hybridisation

Specific primers (5'- ACGTGTTCTTAATTTGCTTTTCCCT-3' and 5'- CCCCTGCGGGTAGGTGAT-3') were designed to amplify exons 4 and 5 of mouse *Rac1* cDNA (the exons deleted in the conditional mutants used [*Glogauer et al., 2003*]), generating a 200 bp fragment. *Nkx1-2* probe was a kind gift from Dr F. Schubert (*Schubert et al., 1995*). Whole-mount in situ hybridisations were performed using digoxigenin-labelled sense and anti-sense RNA probes, followed by preparation of 40 µm vibratome sections.

## Histology

Embryos were fixed overnight in Bouin's solution (Sigma), dehydrated in an ethanol series and embedded in paraffin-wax. Seven micron sections were stained using Harris' haematoxylin solution and 2% Eosin Y (both Sigma). Images were captured on an Axiophot2 upright microscope.

## Scanning electron microscopy

Embryos were fixed overnight in 2% glutaraldehyde, 2% paraformaldehyde in 0.1 M phosphate buffer, pH7.4, at 4°C, post-fixed in 1% $OsO_4$/1.5% $K_4Fe(CN)_6$ in 0.1 M phosphate buffer at 3°C for 1.5 hr and then rinsed in 0.1 M phosphate buffer. After rinsing with distilled water, specimens were dehydrated in a graded ethanol-water series to 100% ethanol, followed by one acetone wash. The samples were then critical point dried using $CO_2$ and mounted on aluminium stubs using sticky carbon taps. The mounted samples were then coated with a thin layer of Au/Pd (approximately 2 nm thick) using a Gatan ion beam coater and imaged with a JEOL 7401 FEGSEM.

## Serial block-face scanning electron microscopy

Embryos were fixed for 12–36 hr in 3% glutaraldehyde and 1% paraformaldehyde in 0.08 M sodium cacodylate buffer, pH 7.4, and then *en bloc* stained with osmium ferricyanide-thiocarbohydrazide-osmium, uranyl acetate, and Walton's lead citrate as described (*West et al., 2010*) with two modifications. First, the osmium concentration was reduced to 1% and, second, graded alcohols (50, 70, 90, 3 x 100%) and propylene oxide were used instead of acetone to dehydrate specimens for infiltration and curing overnight at 60°C in Durcupan ACM resin. Specimens were then superglued to aluminium pins and trimmed to place the region of interest within a 0.5 x 0.5 x 0.4 mm mesa and sputter coated with 5 nm gold palladium. Stacks of backscatter electron micrographs were automatically acquired using a Gatan 3 view system in conjunction with a Zeiss Sigma field emission scanning electron microscope working in variable pressure mode at a chamber pressure of 9 Pa and 4 kV. At a standard magnification of x1000 and a pixel resolution of 4096 x 4096, the total area sampled measured 255.4 µm² on x and y and, depending on the number of 100-nm-thick sections sampled, between 67 and 150 µm on z. The resulting stacks were normalised for contrast and brightness and

then converted to TIFF images in Digital Micrograph prior to importation into Amira 5.3.3 software for semi-automated segmentation and presentation.

## Protrusion analysis

Protrusions were scored based on SEM images of the PNP fusion point taken at 2000x magnification, and categorised in four different classes: <u>**Ruffles**</u> (comprised predominantly or solely of membrane ruffles), <u>**Ruffles**</u> and <u>**Filopodia**</u> (either a mixture of both types of protrusions, filopodia that emanate from ruffles, or ruffles with microspikes), <u>**Filopodia**</u> (comprised predominantly or solely of filopodial protrusions), <u>**Absent**</u> (total absence of protrusions, or just one or two incipient protrusions). Examples of these types of protrusions can be found in *Figure 2—figure supplement 2*, and the full dataset of protrusion images can be found in the Dryad Digital Repository (doi:10.5061/dryad.rm660).

Scoring was done blind to embryonic stage and genotype by two different persons. In the minority of cases where the two scorings did not concur, a final decision was made by consensus.

Where analysed, filopodial density was measured by counting the number of individual filopodia in an area of 2000 $\mu m^2$ around the point of neural fold fusion, and filopodial length was measured in the same area using Fiji software. Only filopodia that measured above 1 $\mu m$ were considered for these analyses.

## Statistical analysis

Kruskal-Wallis ANOVA on ranks was used for comparison of PNP size between different groups within each stage range. Fisher exact test was used for comparison of proportions of different types of protrusions and Chi-square test was used for comparison of phenotype frequencies in *Table 1*; when more than two groups were compared in multiple tests, the alpha-level was protected manually. Mann-Whitney Rank Sum Test was used to compare filopodial number and filopodial length (*Figure 6G,H*).

## Acknowledgements

We thank Kate Storey for sharing unpublished reagents (*Nkx1-2^CreERT2* line), Mark Turmaine for assistance with SEM, Alix Palmer for providing *Sp^2H* embryos, and Maddy Parsons and Juan-Pedro Martinez-Barbera for helpful comments on the manuscript.

## Additional information

### Funding

| Funder | Grant reference number | Author |
| --- | --- | --- |
| Wellcome Trust | 087525 | Andrew J Copp |
| Medical Research Council | G0801124 | Andrew J Copp |
| Medical Research Council | G0802163 | Nicholas DE Greene |

The funders had no role in study design, data collection and interpretation, or the decision to submit the work for publication.

### Author contributions

AR, Conception and design, Acquisition of data, Analysis and interpretation of data, Drafting or revising the article; DS, Conception and design, Acquisition of data, Drafting or revising the article; SE, SCdC, HEJA, Acquisition of data, Analysis and interpretation of data, Drafting or revising the article; PMGM, Analysis and interpretation of data, Drafting or revising the article; MAM, Acquisition of data, Drafting or revising the article; NDEG, AJC, Conception and design, Analysis and interpretation of data, Drafting or revising the article

### Author ORCIDs

Ana Rolo, http://orcid.org/0000-0002-8683-4991
Andrew J Copp, http://orcid.org/0000-0002-2544-9117

### Ethics

Animal experimentation: Animal studies were performed according to the UK Animals (Scientific Procedures) Act 1986 and the Medical Research Council's Responsibility in the Use of Animals for Medical Research (July 1993), under Project Licence number PPL 70/7469 held by A. Copp.

## Additional files

### Major datasets

The following dataset was generated:

| Author(s) | Year | Dataset title | Dataset URL | Database, license, and accessibility information |
|---|---|---|---|---|
| Rolo A, Savery D, Castro SC, Armer HEJ, Munro PMG, Mole MA, Greene NDE, Copp AJ | 2016 | Data from: Regulation of cell protrusions by small GTPases during fusion of the neural folds | http://dx.doi.org/10.5061/dryad.rm660 | Available at Dryad Digital Repository under a CC0 Public Domain Dedication |

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
