## [Decision Letter]

Thank you for submitting your work entitled "Regulation of Cell Protrusions by Small GTPases during Fusion of the Neural Folds" for consideration by *eLife*. Your article has been favorably evaluated by Fiona Watt as Senior editor and three reviewers, one of whom, Marianne Bronner, is a member of our Board of Reviewing Editors.

The reviewers have discussed the reviews with one another and the Reviewing Editor has drafted this decision to help you prepare a revised submission

Summary:

This is an elegant paper that uses a series of mouse mutants to understand the regulation of cell protrusion during fusion of the neural plate. While we have some understanding of the genetics underlying neural tube defects, the cell biological events are not well described. This is in large part due to the difficulties studying extremely dynamic interactions during embryonic stages that are not readily accessible.

The authors demonstrate a role for membrane ruffles in the surface ectodermal cells which requires Rac1, as well as role for Cdc42 in filopodia in the neural folds. While it is unsurprising that these molecules are important in neural tube fusion, the identification of the key cells (surface ectodermal cells) is important and significant. Furthermore, their analysis suggests that these protrusions are necessary during particular stages of neurulation and that the nature of the protrusions changes during development (from predominantly filopodial early to later ruffles and filopodia.

The characterisation of cell protrusions in different neurulation stages, as well as the contribution of Rho GTPases to different protrusions has been carried out carefully and has developmental significance. Many of the observations are novel and interesting. However, several essential controls are missing. Once corrected, this paper would be highly appropriate for *eLife*.

Essential revisions:

1) Regarding the observation that cranial neural tube closure in Rac1/Cdc42 mutants is not as affected as the posterior neural tube, this would be better supported with some comparative data. However, we appreciate that may be beyond the scope of this paper. Therefore, we suggest that the Discussion not close with this section.

2) While the section on epithelial closure is useful and informative, we would have liked a summary of similarities and differences of NTD etiology and membrane ruffling/filopodial dynamics in different organisms (e.g., mouse vs. chick vs. frog).

3) The Pax3^cre^ allele is a heterozygous mutation of *Pax3*, which has a phenotype. The genetics tables are helpful in allaying worries about genetic interactions; however, the authors should still discuss any possible Pax3-dependent cytoskeletal regulation – e.g., Wiggan et al., Cell Signal. 2006 and subsequent papers.

4) Although in Pax3cre-Rac1 and Grhl3cre-Rac1 embryos, the decrease of ruffles is accompanied with posterior neural tube closure defects, it is not certain that loss of ruffles is the cause of the neural tube closure defects. Another important possibility is that Rac1 knockdown results in instability of the actin-Adherens Junctions complex, which is involved in force transmission between adjacent cells to accomplish cell elongation and tissue movement and it is this that impairs active neural tube closure. The authors need to examine the localisation of F-actin and adherens junction components in Rac1 conditional knockouts, before they can reach a conclusion about the role of protrusions in neural tube closure.

5) The mRNA and proteins of Rac1 and Cdc42 are usually stored at a high level in different cell types since they are important for fundamental cellular activities. Therefore, it is arguable if the conditional knockouts, which are driven by the promoters activated in the early or late neurulation stages, can effectively knockdown the proteins immediately. The authors should provide assessment of protein knockdown in the different conditional knockouts and in the different neurulation stages.

6) The authors observed that the protrusive cells have a SE cell like shape and thus concluded that the protrusions come from SE cells. This is not necessary the case as, if NE cells undergo the EMT process, they could also transform into protrusive cells exemplified in neural crest migration. To fully verify cell identify, it would be better to examine cell lineage markers or cell-type specific junction markers. Since surface ectoderm and presumptive neural crest are in such close proximity and neural crest cells do expression some E-cadherin, the distinction seems a bit semantic. The authors should better explain how to distinguish these cell types or be more cautious in their interpretation.

7) This paper is really about cellular protrusions during neurulation, rather than epithelial adhesion and fusion. This does not detract from the work, but it might help the Introduction and Discussion to clarify that point, since the authors never actually look at the "fusion".

---

## [Author Response]

*Essential revisions: 1) Regarding the observation that cranial neural tube closure in Rac1/cdc42 mutants is not as affected as the posterior neural tube, this would be better supported with some comparative data. However, we appreciate that may be beyond the scope of this paper. Therefore, we suggest that the Discussion not close with this section.*

We agree and have now removed this section.

*2) While the section on epithelial closure is useful and informative, we would have liked a summary of similarities and differences of NTD etiology and membrane ruffling/filopodial dynamics in different organisms (e.g., mouse vs. chick vs. frog).*

We have now expanded this section in the Introduction to include details about different taxa (Introduction, third paragraph).

*3) The Pax3^cre^ allele is a heterozygous mutation of Pax3, which has a phenotype. The genetics tables are helpful in allaying worries about genetic interactions; however, the authors should still discuss any possible Pax3-dependent cytoskeletal regulation – e.g., Wiggan et al., Cell Signal. 2006 and subsequent papers.*

We now address this possibility (see subsection “Rac1 is required for the formation of ruffles during late stages of neural tube closure”, first paragraph) and further explore it by looking at protrusions in the *Pax3* mutant *Sp^2H^*. We found no difference between protrusions formed by either wild-type, heterozygous or *Sp^2H^* mutants. We have included these data in the raw data sets deposited with Dryad.

*4) Although in Pax3cre-Rac1 and Grhl3cre-Rac1 embryos, the decrease of ruffles is accompanied with posterior neural tube closure defects, it is not certain that loss of ruffles is the cause of the neural tube closure defects. Another important possibility is that Rac1 knockdown results in instability of the actin-Adherens Junctions complex, which is involved in force transmission between adjacent cells to accomplish cell elongation and tissue movement and it is this that impairs active neural tube closure. The authors need to examine the localisation of F-actin and adherens junction components in Rac1 conditional knockouts, before they can reach a conclusion about the role of protrusions in neural tube closure.*

We have now explored this possibility by looking at the distribution of F-actin, β-catenin and E-cadherin, as suggested by the reviewers. The data and discussion of this new work are in the last paragraph of the subsection “Rac1 is required for the formation of ruffles during late stages of neural tube closure”, and in Figure 3—figure supplement 4. We saw no differences between the distribution of these proteins between Pax3Cre-Rac1 mutant and control embryos, in either the SE or NE cells. Also, the protein distribution in cells of the targeted and untargeted regions of mutant embryos looks similar. These results reinforce the idea that the NTDs observed are very likely the result of the protrusive defect, rather than any epithelial instability, which we cannot detect.

*5) The mRNA and proteins of Rac1 and Cdc42 are usually stored at a high level in different cell types since they are important for fundamental cellular activities. Therefore, it is arguable if the conditional knockouts, which are driven by the promoters activated in the early or late neurulation stages, can effectively knockdown the proteins immediately. The authors should provide assessment of protein knockdown in the different conditional knockouts and in the different neurulation stages.*

We accept this possibility, however we feel it is very likely we achieved sufficient protein knockdown in our study, because:

a) Rac1 protein is actively degraded by cells via the ubiquitin-proteasome – see (Nethe and Hordijk, 2010) for a review with several examples – and the half-life of active Rac1 is of ~3hours (Kovacic et al., 2001), indicating a fast turnover of protein;

b) “Immediate” knockdown of the protein would not be required as long as the promoters driving Cre recombination are activated early enough in development;

c) In Wnt1Cre-Rac1 embryos, Rac1 protein is completely absent from the first pharyngeal arch (almost completely derived from neural crest cells, which express Wnt1Cre) from at least E10.0, possibly earlier (Thomas et al., 2010). Wnt1Cre-driven reporter expression is first detected in the cells that will form the first pharyngeal arch from ss11 (Jacques-Fricke et al., 2012), whereas Pax3Cre-driven expression can be detected in our lines from at least ss5 (see Figure 3), which is about 12 hours earlier than ss11, meaning that by at least E9.5 the protein should be absent from the targeted tissues in our study;

d) We found effective mRNA knock-down of Rac1 in the targeted tissues (Figure 3—figure supplement 2 and Figure 6—figure supplement 1);

e) Pax3Cre-Rac1 and Pax3Cre-Cdc42 embryos both develop fully penetrant neural crest-derivative phenotypes later in development (as shown in Figure 3—figure supplement 4 and Figure 5—figure supplement 1), including at hindlimb levels correspondent to the point of NT closure at E9.5. This reinforces that the proteins were successfully targeted at all the rostro-caudal levels.

Nonetheless, we have made several attempts at detecting Rac1 protein by immunofluorescence in PNP sections of E9.5 embryos. However, after being unable to find in the literature any convincing examples of such detection at stages earlier than E11, and in view of the insensitivity of the antibodies we have tried, we concluded that this approach would be unfeasible and that protein reduction could probably only be shown by western blotting. Given that we are dealing with conditional KOs, not all cells in an embryonic structure (e.g. caudal embryonic region) would be expected to show protein knockdown. Moreover, it would be unfeasible to precisely dissect out cells of the Pax3Cre domain. In a previous study using qRT-PCR on dissected PNP regions of E9.5 embryos (the same region we would analyse) (Zhao et al., 2014), conditional KO of β-catenin using the Pax3Cre driver led to a reduction of only about 20% of mRNA levels. Therefore, an approach to this question using western blotting would suffer from the need to detect knockdown of Rac1 in a minority of cells within the dissected samples. We consider that the large number of embryos required to have a chance of detecting such small levels of protein reduction in the different conditional knockouts, and at the different neurulation stages, would be prohibitive in terms of cost and time.

Erickson, C.A., and Weston, J.A. (1983). An SEM analysis of neural crest migration in the mouse. Journal of embryology and experimental morphology 74, 97-118.

Jacques-Fricke, B.T., Roffers-Agarwal, J., and Gammill, L.S. (2012). DNA methyltransferase 3b is dispensable for mouse neural crest development. PLoS One 7, e47794.

Kovacic, H.N., Irani, K., and Goldschmidt-Clermont, P.J. (2001). Redox regulation of human Rac1 stability by the proteasome in human aortic endothelial cells. The Journal of biological chemistry 276, 45856-45861.

Nethe, M., and Hordijk, P.L. (2010). The role of ubiquitylation and degradation in RhoGTPase signalling. Journal of cell science 123, 4011-4018.

Thomas, P.S., Kim, J., Nunez, S., Glogauer, M., and Kaartinen, V. (2010). Neural crest cell-specific deletion of Rac1 results in defective cell-matrix interactions and severe craniofacial and cardiovascular malformations. Developmental biology 340, 613-625.

Zhao, T., Gan, Q., Stokes, A., Lassiter, R.N., Wang, Y., Chan, J., Han, J.X., Pleasure, D.E., Epstein, J.A., and Zhou, C.J. (2014). beta-catenin regulates Pax3 and Cdx2 for caudal neural tube closure and elongation. Development 141, 148-157.

*6) The authors observed that the protrusive cells have a SE cell like shape and thus concluded that the protrusions come from SE cells. This is not necessary the case as, if NE cells undergo the EMT process, they could also transform into protrusive cells exemplified in neural crest migration. To fully verify cell identify, it would be better to examine cell lineage markers or cell-type specific junction markers. Since surface ectoderm and presumptive neural crest are in such close proximity and neural crest cells do expression some E-cadherin, the distinction seems a bit semantic. The authors should better explain how to distinguish these cell types or be more cautious in their interpretation.*

We accept the point that morphology alone is not sufficient to determine cell type of origin. We would like to point out, however, that the prospective neural crest region, at the axial level we have analysed, is pre-migratory. Neural crest cells do not migrate out of the spinal region until at least 12 hours after neural tube closure is completed (Erickson and Weston, 1983), therefore making it unlikely that cells at the point of neural tube closure have yet begun to undergo EMT. Nonetheless, we accept that these could be highly modified NE cells and have now changed our interpretation accordingly at the end of the Results section.

Erickson, C.A., and Weston, J.A. (1983). An SEM analysis of neural crest migration in the mouse. Journal of embryology and experimental morphology 74, 97-118.

*7) This paper is really about cellular protrusions during neurulation, rather than epithelial adhesion and fusion. This does not detract from the work, but it might help the Introduction and Discussion to clarify that point, since the authors never actually look at the "fusion".*

Yes, in fact this paper is about the cellular protrusions that occur before the actual fusion occurs. We have clarified this point by changing the text in the Abstract and Introduction (third and fifth paragraphs), emphasizing that cell protrusions occur just prior to neural fold fusion.